# The differentiation state of the Schwann cell progenitor drives phenotypic variation between two contagious cancers

**Rachel S. Owen**[1], **Sri H. Ramarathinam**[2], **Alistair Bailey**[3,4], **Annalisa Gastaldello**[1], **Kathryn Hussey**[1], **Paul J. Skipp**[1,4], **Anthony W. Purcell**[2], **Hannah V. Siddle**[1,4]*

**1** School of Biological Sciences, University of Southampton, Southampton, United Kingdom, **2** Department of Biochemistry and Molecular Biology and the Infection and Immunity Program, Biomedicine Discovery Institute, Monash University, Clayton, Australia, **3** Centre for Cancer Immunology, University of Southampton, Southampton, United Kingdom, **4** Institute for Life Sciences, University of Southampton, Southampton, United Kingdom

* H.V.Siddle@soton.ac.uk

## Abstract

Contagious cancers are a rare pathogenic phenomenon in which cancer cells gain the ability to spread between genetically distinct hosts. Nine examples have been identified across marine bivalves, dogs and Tasmanian devils, but the Tasmanian devil is the only mammalian species known to have given rise to two distinct lineages of contagious cancer, termed Devil Facial Tumour 1 (DFT1) and 2 (DFT2). Remarkably, DFT1 and DFT2 arose independently from the same cell type, a Schwann cell, and while their ultra-structural features are highly similar they exhibit variation in their mutational signatures and infection dynamics. As such, DFT1 and DFT2 provide a unique framework for investigating how a common progenitor cell can give rise to distinct contagious cancers. Using a proteomics approach, we show that DFT1 and DFT2 are derived from Schwann cells in different differentiation states, with DFT2 carrying a molecular signature of a less well differentiated Schwann cell. Under inflammatory signals DFT1 and DFT2 have different gene expression profiles, most notably involving Schwann cell markers of differentiation, reflecting the influence of their distinct origins. Further, DFT2 cells express immune cell markers typically expressed during nerve repair, consistent with an ability to manipulate their extracellular environment, facilitating the cell's ability to transmit between individuals. The emergence of two contagious cancers in the Tasmanian devil suggests that the inherent plasticity of Schwann cells confers a vulnerability to the formation of contagious cancers.

## Author summary

Cancer can be an infectious pathogen, with nine known cases, infecting bivalves, dogs and two independent tumours circulating in the endangered Tasmanian devil. These cancers, known as Devil Facial Tumour 1 (DFT1) and Devil Facial Tumour 2 (DFT2), spread through the wild population much like parasites, moving between genetically distinct

**Data Availability Statement:** The mass spectrometry proteomics data have been deposited to the ProteomeXchange Consortium via the

PRIDE partner repository with the dataset identifier PXD021784. All other relevant data is available in the manuscript and/or supporting information files. Supplementary tables used to generate figures in the paper can be also found at https://doi.org/10.5258/SOTON/D1975.

**Funding:** This work was funded by a Leverhulme Trust project grant (RPG-2015-103) to HVS and AWP (leverhulme.ac.uk/). AB is funded by a Cancer Research UK Centres Network Accelerator Award, A21998 (cancerresearchuk.org) and RO is funded by a Morris Animal Foundation Fellowship, D19ZO-413 (morrisanimalfoundation.org/). Instrumentation in the Centre for Proteomic Research, University of Southampton is supported by the BBSRC (BM/M012387/1; bbsrc.ukri.org/) to PJS, and the Wessex Medical Trust. AWP is supported by a NHMRC Principal Research Fellowship, 1137739 (nhmrc.gov.au). Monash Proteomics & Metabolomics Facility provided mass spectrometry instrumentation and technical support. The funders had no role in study design, data collection and analysis, decision to publish, or preparation of the manuscript.

**Competing interests:** The authors have declared that no competing interests exist.

hosts during social biting behaviours and persisting in the population. As DFT1 and DFT2 are independent contagious cancers that arose from the same cell type, a Schwann cell, they provide a unique model system for studying the emergence of phenotypic variation in cancers derived from a single progenitor cell. In this study, we have shown that these two remarkably similar tumours have emerged from Schwann cells in different differentiation states. The differentiation state of the progenitor has altered the characteristics of each tumour, resulting in different responses to external signals. This work demonstrates that the cellular origin of infection can direct the phenotype of a contagious cancer and how it responds to signals from the host environment. Further, the plasticity of Schwann cells may make these cells more prone to forming contagious cancers, raising the possibility that further parasitic cancers could emerge from this cell type.

## Introduction

Identifying the origin of a pathogen is key to predicting its evolutionary trajectory, the mechanisms of emergence and developing appropriate disease management strategies. Contagious cancers can emerge when malignant cells gain the ability to spread between genetically distinct hosts and propagate through a population in a manner akin to a parasite [1]. These cancers present a significant health risk to their hosts and a conservation issue when circulating in vulnerable populations. How contagious cancers emerge is poorly understood, but their origins and evolution can inform on the emergence and evolution of cancers more generally.

The Tasmanian devil is the only mammalian species harbouring two lineages of contagious cancer within its population. Devil Facial Tumour 1 (DFT1 or DFTD) emerged in the 1990s [2] and has caused significant population declines over the last 25 years [3]. DFT1 presents as large tumours around the face and oral cavity [4] which are strongly positive for the myelin associated marker protein Periaxin (PRX) [5]. In 2014, grossly identical facial tumours were identified which have a distinct histology, which lacked expression of PRX [6]. This new tumour was termed DFT2, and its prevalence and range is increasing [7]. All evidence indicates that DFT1 and DFT2 are genetically distinct tumours which have emerged independently of one another [6,8].

Remarkably, both DFT1 and DFT2 have emerged from the same cell type, a Schwann cell [9,10]. Schwann cells are the principle glial cell type of the peripheral nervous system which provide protection and trophic support to axons [11], produce the myelin sheath to aid nerve conduction [12] and are key to nerve regeneration after injury [13]. Schwann cells develop from the neural crest through intermediate differentiation states, from stem-like Schwann cell precursors to Immature Schwann cells and finally to the terminally differentiated Myelinating and Non-Myelinating Schwann cells, which show significantly reduced proliferation and motility compared to less well differentiated Immature Schwann cells [14,15]. Terminally differentiated Schwann cells retain the ability to rapidly de-differentiate to a Repair Schwann cell in response to nerve injury [16], a differentiation state which shares many similarities with Immature Schwann cells but has a distinct phenotype [17]. There is evidence that cellular plasticity can predispose cells to an increased risk of neoplastic transformation [18] and Schwann cell derived tumours in humans have potential for either benign (Schwannoma) or malignant (Malignant Peripheral Nerve Sheath Tumour) growth [19]. Contagious cancers have only been identified in two mammalian species, with only one other known lineage affecting dogs [20]. Thus, the emergence of two such cancers from Schwann cells in a single species was

unexpected and may indicate a cell type specific vulnerability to the formation of contagious cancers.

The independent emergence of two contagious tumours from a Schwann cell poses a unique opportunity to study the underlying causes of significant phenotypic variation seen in Schwann cell tumours afflicting other species [21]. These tumours can range from largely benign, well encapsulated Schwannomas [22] through to aggressive, metastatic Malignant Peripheral Nerve Sheath Tumours (MPNST) which are often resistant to treatment and demonstrate a high frequency of recurrence in patients [23]. At least some of this variability is thought to derive from the differentiation state of the Schwann cell progenitor at the point of cancerous transformation [24,25], with more aggressive and metastatic cancers associated with less differentiated progenitor cells. The effects of the progenitor cell characteristics on tumour phenotype additionally extends to oncovirus-associated cancers [26]. While no viral origin for DFTs has been identified [8], oncovirus associated cancers are a relatively poorly understood area of conservation science, and represent a significant and widespread problem in many wild populations [27]. Despite their shared origins, DFT1 and DFT2 are distinct tumours, exhibiting variation in immune evasion mechanisms [28], mutational landscape [8] and enrichment for myelination pathways [10]. Thus, these tumours provide an opportunity to study how inherent features of the progenitor cell direct tumour phenotype, particularly with regards to immune evasion.

In this study we have investigated the origins of the histological and phenotypic diversity demonstrated by DFT1 and DFT2. We show that DFT2 emerged from a less well differentiated Schwann cell than DFT1, and that these divergent cellular origins have resulted in distinct responses to inflammation, directing the emergence of immune evasive phenotypes in each tumour. The data presented indicates that the emergence of DFT1 and DFT2 may have been facilitated by the inherent plasticity and adaptability of Schwann cells, and that the differentiation state of a Schwann cell progenitor can have significant and wide reaching effects on the phenotype of the cancer it forms.

## Results

### DFT2 emerged from a less differentiated Schwann cell than DFT1

Whole cell proteomes were generated in triplicate for representative cell lines derived from DFT1 (DFT1_4906), DFT2 (DFT2_RV) and immortalised devil fibroblasts (Fibs_Salem). In this dataset 3874 common proteins identified in all three replicates from each cell line were quantified (Fig 1A). The most highly overexpressed proteins in DFT1 relative to fibroblasts includes the myelin associated proteins PRX[29] (9.97 fold, $p = 6.52$ x$10^{-7}$) and Myelin Protein Zero (MPZ) [30] (9.06 fold, $p = 5.34$ x$10^{-5}$) (Fig 1B). Other myelin associated proteins significantly overexpressed in DFT1 relative to fibroblasts include Early Growth Response protein 2 (EGR2)[31] (2.10 fold, $p = 4.67$ x$10^{-3}$), Myelin Associated Glycoprotein (MAG) [32] (5.56 fold, $p = 3.51$ x$10^{-6}$) and Uridine Diphosphate Glycosyltransferase 8 (UGT8) [33] (4.74 fold, $p = 1.78$ x$10^{-4}$) (S1 Table). These proteins are consistent with previous findings that DFT1 emerged from a well differentiated myelinating Schwann cell [9,34].

In contrast to DFT1, protein expression in DFT2 indicates a less differentiated cell type, albeit with some Schwann cell markers. The two highest overexpressed proteins in DFT2 relative to fibroblasts were L1 Cell Adhesion Molecule (L1CAM) (8.35 fold, $p = 8.80$ x$10^{-5}$) and Nerve Growth Factor Receptor (NGFR) (8.14 fold, $p = 1.47$ x$10^{-4}$), both of which are associated with immature and repair Schwann cells [35,36] (Fig 1C). Other proteins significantly overexpressed in DFT2 relative to fibroblasts include the immature glial cell marker Fatty Acid Binding Protein 7 (FABP7) [37] (7.07 fold, $p = 8.11$ x$10^{-4}$), nerve specific Microtubule Associated

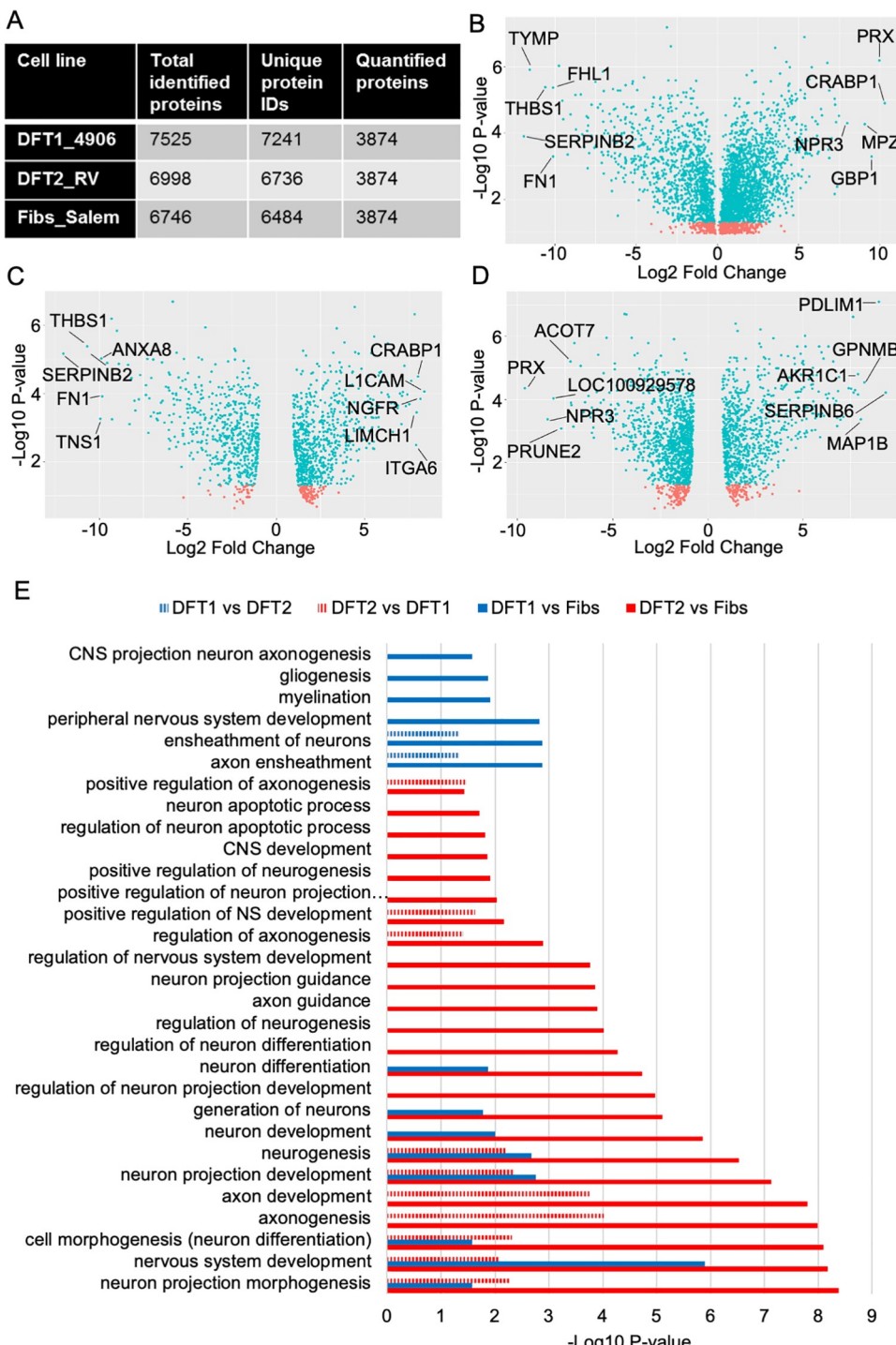

**Fig 1. The DFT2 proteome is enriched for proteins indicative of a de-differentiated Schwann cell.** A) Summary table indicating the number of unique proteins identified in three independent biological replicates of each cell line. B-D) Volcano plots indicating the differential expression of proteins between B) DFT1_4906 and Fibs_Salem, where the right half of the graph indicates proteins overexpressed in DFT1 relative to fibroblasts, C) DFT2_RV and Fibs_Salem, where the right half of the graph indicates proteins overexpressed in DFT2 relative to fibroblasts and D) DFT2_RV and DFT1_4906. The right half of each graph indicates proteins overexpressed in DFT2 relative to DFT1. The X axis represents positive or negative Log2 fold change of protein expression between two cell lines, Y axis represents T-test p-value determining the significance of differential expression. Blue points indicate proteins which are significantly differentially expressed and red points indicate proteins whose differential expression is not

statistically significant. The top 5 most highly over and under-expressed proteins are labelled. Pairwise comparison data used to generate these plots is presented in S1–S3 Tables. E) Visual representation of all nervous system related biological processes enriched in the proteome. Overexpressed proteins (>2 fold) of DFT2_RV relative to Fibs_Salem (DFT2 vs Fibs, red), DFT1_4906 relative to Fibs_Salem (DFT1 vs Fibs, blue), DFT2_RV relative to DFT1_4906 (DFT2 vs DFT1, red dashed) and DFT1_4906 relative to DFT2_RV (DFT1 vs DFT2, blue dashed) are shown. Enrichment data was generated using gProfiler and nervous system associated processes were manually identified. X axis represents the significance of biological pathway enrichment. NS = Nervous system. CNS = Central nervous system. Full list of enriched biological processes identified in each protein set are presented in S4–S7 Tables.

Protein 2 (MAP2) [38] (5.17 fold, $p = 1.20 \times 10^{-4}$) and pan-Schwann cell lineage marker S100 Calcium Binding Protein B (S100B)[39] (2.66 fold, $p = 1.75 \times 10^{-3}$) (S2 Table), indicating that DFT2 has emerged from a less differentiated, immature or repair Schwann cell.

Pairwise comparisons between DFT1 and DFT2 reveals that structural myelin proteins are more highly expressed in DFT1 than DFT2. The most highly overexpressed protein in DFT1 relative to DFT2 is PRX [5] (9.61 fold, $p = 4.44 \times 10^{-5}$) (Fig 1D). Other structural myelin proteins, including MPZ (6.10 fold, $p = 1.17 \times 10^{-3}$) and MAG (5.77 fold, $p = 3.88 \times 10^{-6}$) are overexpressed by DFT1 versus DFT2 (S3 Table), whilst neuron and repair Schwann cell marker Microtubule Associated Protein 1B (MAP1B) [40] (4.60 fold, $p = 4.25 \times 10^{-4}$) and immature glial markers L1CAM (7.88 fold, $p = 1.62 \times 10^{-5}$) and Platelet Derived Growth Factor Receptor Alpha (PDGFRA) [41] (7.71 fold, $p = 5.15 \times 10^{-5}$) are amongst the most highly overexpressed proteins in DFT2 relative to DFT1 (Fig 1D and S3 Table).

Next the biological pathways enriched in each tumour relative to fibroblasts were identified. Proteins which were significantly overexpressed ≥ 2 fold in DFT1 or DFT2 compared to fibroblasts were analysed in gProfiler as a ranked list from highest to lowest expression (Fig 1E). Both tumours demonstrate significant enrichment of biological processes associated with the nervous system and Schwann cells including "nervous system development" (DFT1: $p = 1.28 \times 10^{-6}$. DFT2: $p = 4.45 \times 10^{-9}$) and "neuron differentiation" (DFT1: $p = 1.34 \times 10^{-2}$. DFT2: $p = 3.80 \times 10^{-5}$). However, DFT1 is enriched for the Schwann cell specific processes of "myelination" ($p = 1.22 \times 10^{-2}$), "axon ensheathment" ($p = 1.32 \times 10^{-3}$) and "ensheathment of neurons" ($p = 1.32 \times 10^{-3}$) (Fig 1E, blue solid), whilst DFT2 demonstrates unique enrichment for additional biological processes associated with the development and regulation of the nervous system, including "central nervous system development" ($p = 4.04 \times 10^{-2}$), "axonogenesis" ($p = 1.12 \times 10^{-8}$) and "regulation of neuron differentiation" ($p = 2.49 \times 10^{-4}$) (Fig 1E, red solid). Proteins which were significantly overexpressed ≥ 2 fold in DFT1 relative to DFT2 (DFT1 vs DFT2) or in DFT2 relative to DFT1 (DFT2 vs DFT1) were analysed in the same way and show that DFT1 is significantly enriched for "axon ensheathment" ($p = 4.89 \times 10^{-2}$) and "ensheathment of neurons" ($p = 4.89 \times 10^{-2}$) relative to DFT2 (Fig 1E, blue dashed), whilst DFT2 is significantly enriched for nervous system developmental processes relative to DFT1 including "neuron projection morphogenesis" ($p = 5.45 \times 10^{-3}$) and "axon development" ($p = 9.46 \times 10^{-5}$) (Fig 1E, red dashed) indicating that DFT1 has emerged from a more well differentiated Schwann cell than DFT2. DFT2 is additionally enriched for biological processes involved in cell motility including "cell migration" ($p = 8.12 \times 10^{-6}$), "cell motility" ($p = 8.04 \times 10^{-5}$) and "locomotion" ($p = 2.03 \times 10^{-5}$), and injury related responses "wound healing" ($p = 1.64 \times 10^{-2}$) and "response to wounding" ($p = 1.25 \times 10^{-2}$) relative to DFT1 (S6 Table), consistent with an immature-like repair differentiation state.

## DFT1 cells are more responsive than DFT2 cells to inflammation

Schwann cells in eutherian mammals are responsive to inflammatory conditions, de-differentiating to a phenotype able to respond to pathogen threats and wound healing [42,43]. We

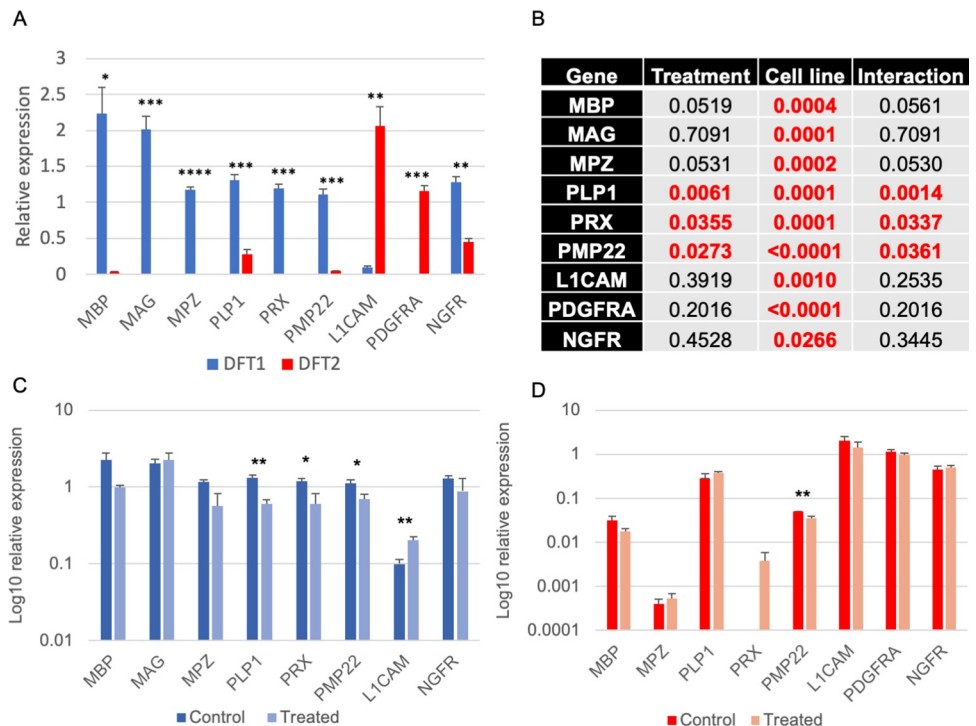

**Fig 2. DFT1 and DFT2 differentially regulate Schwann cell differentiation markers in response to IFNγ treatment *in vitro*.** A) Relative expression of Schwann cell marker genes in DFT1_4906 (blue) and DFT2_RV (red) cells as measured by qPCR. Gene expression was quantified by the relative standard curve method, using RPL13A as a housekeeping control gene. Error bars are SEM. Significance of differential expression as determined by T-test is indicated by stars (* = $P < 0.05$, ** = $P < 0.01$, *** = $P < 0.001$, **** = $P < 0.0001$). B) Two-way ANOVA analysis of the response of DFT1 and DFT2 cells to IFNγ at Schwann cell markers, indicating the significance p-values of differential expression between cell lines, treatments and the interaction between the two. A $P < 0.05$ indicates a significant result, denoted by bold red text. C-D) Relative expression of Schwann cell marker genes as measured by qRT-PCR in C) DFT1_4906 and D) DFT2_RV cells with and without IFNγ treatment. Gene expression was quantified by the relative standard curve method, using RPL13A as a housekeeping control gene. Error bars are SEM. Note that PDGFRA was not detected in DFT1 cells and MAG was not detected in DFT2 cells. Significance of differential expression as determined by T-test is indicated by stars (* = $P < 0.05$, ** = $P < 0.01$). qPCR data used to generate Fig 2 and can be found in S9 Table.

sought to assess the response of DFT1 and DFT2 to the inflammatory cytokine, IFNγ. DFT1_4906 and DFT2_RV cells were treated with recombinant devil IFNγ for 16 hours in triplicate and the expression of Schwann cell markers was analysed by qRT-PCR. Prior to IFNγ treatment, DFT1 cells express significantly higher levels of transcripts from genes associated with myelination (MBP, MAG, MPZ, PLP1, PRX, PMP22) than DFT2 (Fig 2A), consistent with the proteomics data, whilst DFT2 cells express significantly higher levels of the immature and repair Schwann cell marker genes PDGFRA and L1CAM. Two-way ANOVA analysis indicates a statistically significant effect of cell line on the expression of all Schwann cell marker genes analysed (Fig 2B). There is also a statistically significant effect of treatment condition on the expression of myelin associated genes Proteolipid Protein 1 (PLP1), PRX and Peripheral Myelin Protein 22 (PMP22), and a statistically significant interaction between cell line and treatment for these genes ($P = 0.014$, 0.034 and 0.036 respectively (Fig 2B), indicating that the response of these genes to IFNγ treatment is determined by the cell type.

The expression of genes with a role in myelination varies in DFT1 but not DFT2 cells when treated with IFNγ. In DFT1, the expression of four genes (PLP1, PRX, PMP22 and L1CAM) was significantly altered following treatment with IFNγ ($P = 0.0035$, 0.032, 0.030 and 0.009

respectively), with the expression of the myelin associated genes decreasing following IFNγ treatment and the expression of immature and repair Schwann cell marker L1CAM increasing following IFNγ treatment (Fig 2C). In addition, the expression of myelin associated markers Myelin Basic Protein (MBP) and MPZ decreases after IFNγ treatment in DFT1, but the change is not statistically significant ($P$ = 0.074 and 0.147).

In contrast, only PMP22 is significantly differentially expressed in DFT2 cells following IFNγ treatment (Fig 2D), demonstrating a decrease in expression following treatment ($P$ = 0.009). Interestingly, transcripts of PRX, a gene which is not normally detected in DFT2_RV cells, were detected in two replicates of DFT2 following IFNγ treatment, and not detected in the controls, but this was not a statistically significant expression shift ($P$ = 0.065).

## DFT2 cells have different immune evasion mechanisms to DFT1 cells

Fully differentiated Schwann cells in eutherian mammals have low expression of the immune ligands, Major Histocompatibility Complex Class I (MHC-I), to reduce the risk of damaging immune responses [44,45]. It has been previously shown that while DFT2 cells express MHC-I molecules [28], DFT1 cells are negative for MHC-I [46], but that this is reversed in response to IFNγ [46]. Here we show that IFNγ can upregulate cell surface MHC-I in three DFT2 cell lines (Fig 3A). This upregulation is significantly higher in all three DFT2 cell lines than demonstrated in DFT1 (Fig 3B). Classical MHC-I heavy chains (UABC) and non-classical heavy chain *Saha-UK* (UK) transcripts are both upregulated in DFT1 and DFT2, as are transcripts for Beta 2 microglobulin ($\beta_2$-m), a key component of functional MHC-I molecules[47] (Fig 3C). Poly-comb Repressive Complex 2 (PRC2) has been identified as a key driver of MHC-I loss in DFT1 cells [48]. We sought to identify whether PRC2 was also regulating MHC-I expression in DFT2. Western blot analysis indicates that DFT2_RV cells express no EED protein, a key component of PRC2, indicating that MHC-I expression is not modulated by PRC2 in DFT2 (Fig 3D).

There are also differences in the expression of immune associated genes in DFT1 and DFT2 before and after IFNγ treatment (Fig 3E). PD-L1 has previously been shown to be expressed on DFT1_4906 cells following IFNγ treatment [49], similarly we have shown that it is not constitutively expressed but is upregulated by IFNγ treatment in DFT2_RV. DFT2 cells express comparable levels of STAT3 to DFT1, which do not change upon treatment with IFNγ in either cell line, and lower levels of Interleukin 1 Receptor, Type 1 (IL1R1), which is upregulated by both tumours in response to IFNγ. Interestingly, DFT2_RV shows constitutive transcription of immunosuppressive cytokines TGFβ1 and TGFβ2 which are unaffected by treatment with IFNγ. In contrast, DFT1 expresses very low transcript levels of TGFβ1 and TGFβ2 pre-IFNγ.

Given the differences in MHC-I expression between the tumours we sought to assess how similar DFT1 and DFT2 cells are to Tasmanian devil Schwann cells *in vivo* at MHC-I ligands. Tasmanian devil sciatic nerve tissue, highly enriched for Schwann cells (identified by PRX staining), has low expression of classical MHC-I ligands, with most cells showing little or no expression and the positive cells showing largely cytoplasmic staining (Fig 4A). In contrast, most cells within the sciatic nerve stain strongly for a non-classical MHC-I, Saha-UK, with strong membrane localised staining associated with the Schwann cells surrounding each nerve bundle (Fig 4B). DFT1 tumour cells are negative for classical MHC-I and the non-classical MHC-I, Saha-UK (Fig 4C and 4D), whilst DFT2 tumours can be positive or negative for both classical MHC-I and non-classical, Saha-UK (Fig 4E and 4H).

## Discussion

The independent emergence of DFT1 and DFT2 from Schwann cells provides a unique opportunity to define how the characteristics of a progenitor cell can shape the phenotype and

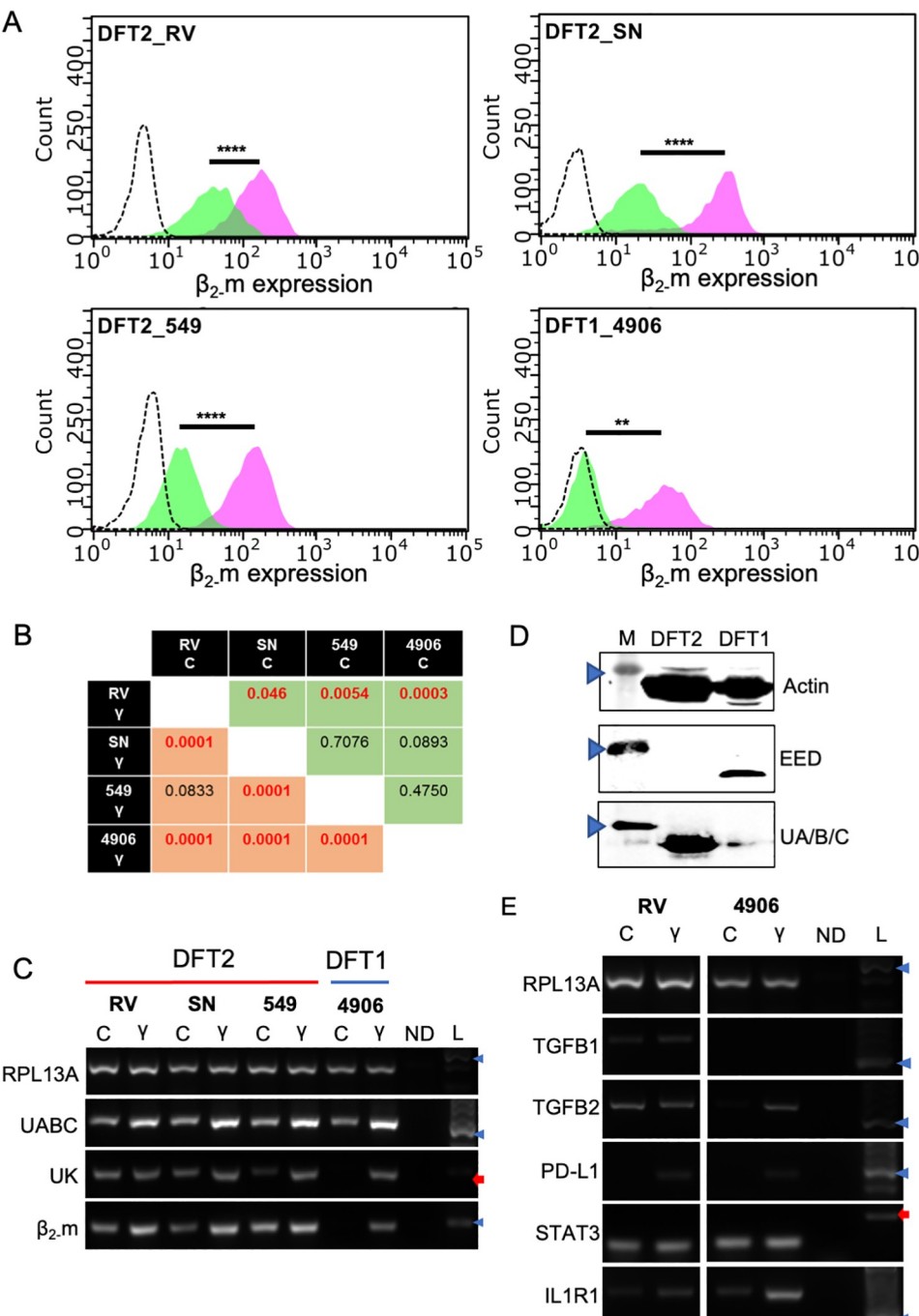

**Fig 3. DFT2 cells have different immune evasion mechanisms to DFT1 cells.** A) Representative flow cytometry histograms demonstrating upregulation of cell surface $\beta_2$_m in response to IFN$\gamma$ treatment in three DFT2 cell lines and one DFT1 cell line. Y-axis indicates cell count, X-axis indicates Log10 fluorescence in the green channel, representing cell surface $\beta_2$_m. Green peaks indicate untreated cells, pink peaks indicate cells treated for 16 hours with recombinant devil IFN$\gamma$, dashed peak indicates secondary antibody only technical control. Data was analysed using using two-way ANOVA and a post-hoc Sidak's multiple comparison test using the Mean Fluorescence Intensity (MFI) of three independent biological replicates for each treatment condition (** = $P < 0.01$, **** = $P < 0.0001$. B) The statistical significance of differential expression of cell surface $\beta_2$_m before and after IFN$\gamma$ treatment. Green boxes indicate comparisons between untreated cell lines. Red boxes indicate comparisons between cell lines following IFN$\gamma$ treatment. Red text indicates statistically significant differences. Data was analysed using two-way ANOVA followed by a post-hoc Tukey's multiple comparison test. C) RT-PCR for MHC class I associated genes on DFT1 and DFT2 following treatment with IFN$\gamma$ in vitro. DFT1 and DFT2 cell lines are indicated on the image. C indicates control cells

grown in normal cell culture media, γ indicates cells treated for 16 hours with recombinant devil IFNγ. ND indicates a no cDNA negative control. L indicates a DNA ladder containing fragments of known size. Key molecular weights are indicated by blue arrowheads (300 bp) and red arrows (200 bp). RPL13A is a housekeeping gene used to control for the amount of cDNA in each PCR reaction. Independent biological replicates of IFNγ treatment were carried out and are presented in S1 Fig. D) Western blots showing the expression of EED protein, a crucial subunit of PRC2, and classical MHC-I heavy chain protein (UABC) in DFT1_4906 and DFT2_RV cells. M indicates a molecular marker of known mass, and blue arrowheads indicate the marker representing ~48 kDa. Beta-Actin is used as a protein loading control. E) RT-PCR for a panel of genes associated with immune regulation in DFT1_4906 and DFT2_RV cell lines following treatment with recombinant devil IFNγ. C indicates control cells grown in normal cell culture media, γ indicates cells treated for 16 hours with recombinant devil IFNγ. ND indicates a no cDNA negative control. L indicates a DNA ladder containing fragments of known size. Key molecular weights are indicated by blue arrowheads (300 bp) and red arrows (200 bp). RPL13A is a housekeeping gene used to control for the amount of cDNA in each PCR reaction. Independent biological replicates of IFNγ treatment were carried out and are presented in S2 Fig.

evolution of an infectious cancer, a key step in predicting the trajectory of pathogen evolution throughout the population. Our results show that although DFT1 and DFT2 both arose from a Schwann cell they have distinct molecular characteristics, which indicates that DFT2 arose from a more de-differentiated Schwann cell.

Patchett *et al.*, (2020) used proteomic and transcriptomic data from two representative cell lines, DFT1_C5065 and DFT2_SN, to demonstrate that DFT2, like DFT1, has emerged from a Schwann cell. Our data supports this finding, indicating consistency between cell lines and validating previous evidence that DFT2 has indeed emerged from a myelinating glial lineage [8,10] (Fig 1). In this study we show that DFT1 expresses high levels of well-differentiated,

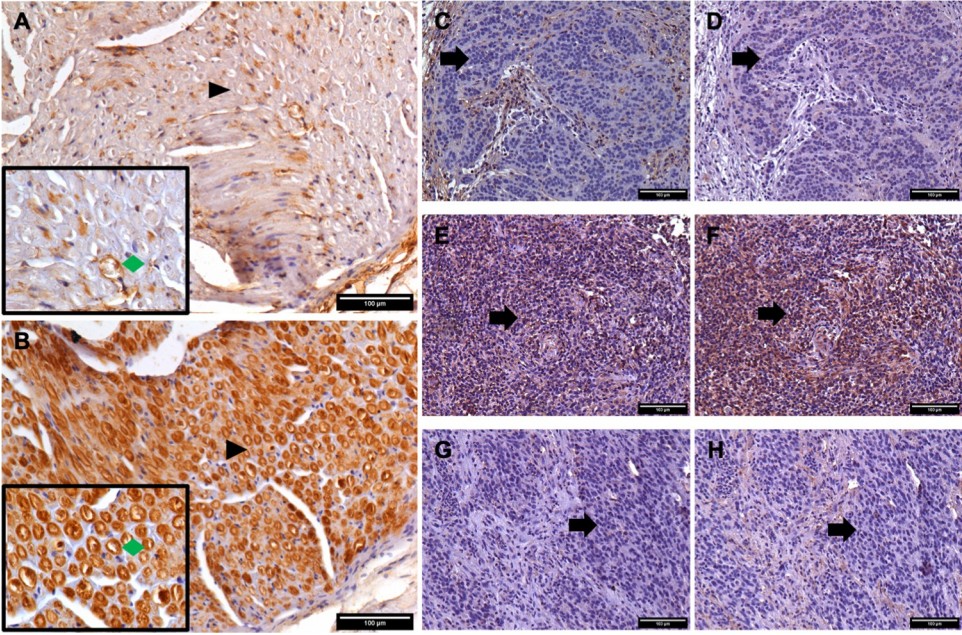

**Fig 4. Tasmanian devil Schwann cells express little classical MHC-I heavy chain and high levels of a non-classical MHC-I heavy chain Saha-UK *in vivo*.** All main images taken at 200X magnification. Scale bars indicate 100 µM. Arrowheads indicate regions imaged at higher magnification. Green diamonds indicate Schwann cells associated with a nerve bundle. Arrows indicate tumour cells. Secondary antibody only controls are included in Supplementary S3 Fig. Brown staining indicates positive staining for the indicated antibody, blue staining indicates cell nuclei counterstained with haematoxylin. A-B) Tasmanian devil sciatic nerve serial sections stained by IHC for A) classical MHC-I heavy chains (UABC) and B) a non-classical MHC-I heavy chain Saha-UK. Boxed images taken at 400X magnification. Arrowheads indicate the region in the boxed image. C-D) DFT1 tumour (Falestinya T1) serial sections stained by IHC for C) UABC and D) Saha-UK. E-F) DFT2 tumour (812 T1) serial sections stained by IHC for E) UABC and F) Saha-UK. G-H-G) DFT2 tumour (547 T1) serial sections stained by IHC for H) UABC and G) Saha-UK.

myelin associated Schwann cell markers including MBP [50], MPZ [30] and MAG [32] and is enriched for biological processes specific to myelination (Figs 1 and 2A). Myelin production in Schwann cells is associated with growth cessation and exit from the cell cycle [14,15], so the expression of several markers of well differentiated myelinating Schwann cells by DFT1 is unusual and unlikely to provide any competitive advantage to tumour cells, indicating their expression is likely carried over from the DFT1 progenitor cell rather than a result of tumour evolution.

In contrast, DFT2 demonstrates low expression of myelin associated markers, and is instead enriched for markers of precursor and immature Schwann cells including FABP7 [37], L1CAM [36] and NGFR [51]. DFT2 cells are also enriched for biological processes contributing to nervous system development, differentiation and regulation (Figs 1 and 2A). This indicates that DFT2 likely emerged from an immature or immature-like repair Schwann cell, whilst DFT1 has likely emerged from a more differentiated myelinating or pro-myelinating Schwann cell. While we cannot completely disregard the possibility that DFT2 emerged from a more well differentiated Schwann cell which has gained a more stem-like phenotype as the tumour has evolved, this hypothesis is challenged by the previously described expression of MHC-I by DFT2 tumours *in vivo* [28], a phenotype not consistent with mature myelinating Schwann cells (Fig 4A) [52] or evolutionarily beneficial to a tumour which relies on avoidance of an allogeneic immune response on transmission to genetically distinct hosts [53,54].

It is possible that DFT2 has emerged from an immature or a repair Schwann cell phenotype and, as has been previously suggested, DFTs may have emerged due to repeated wounding of Schwann cells in the highly innervated whiskers of devils during social biting behaviour [8,10,55]. Given that immature Schwann cells are a developmental state which are not widely seen in adult nerves [56,57], our data suggest that DFT2s emergence from a repair Schwann cell following nerve injury is a likely mechanism of tumour emergence. Some of the difficulty in delineating the specific origin of DFT2 lies in the use of cell lines as opposed to primary tumour samples and the absence of species-specific antibodies. As additional biopsies and reagents for interrogating protein expression in Tasmanian devils become available, it may be possible to further define these origins.

We have demonstrated that DFT2 is less responsive to IFNγ than DFT1 at markers of Schwann cell differentiation, consistent with an already dedifferentiated cell (Fig 2D). DFT1 cells show a reduction in expression of three myelin associated genes following IFNγ treatment, and a significant increase in the expression of L1CAM (Fig 2C), a cell adhesion molecule known to be upregulated by Schwann cells following nerve injury to aid in repair and regeneration [58]. These data indicate that in response to inflammation, DFT1 cells cease myelin production and dedifferentiate to a repair-like phenotype typical of the response to inflammation following nerve injury [59–62]. Our data is consistent with recent evidence that DFT1 tumours dedifferentiate to a stem-like, mesenchymal phenotype in response to vaccine induced immune pressure [63], highlighting the evolutionary benefit that Schwann cell plasticity confers to contagious cancers as they spread into new hosts. In contrast, only a single gene is differentially regulated by DFT2 in response to IFNγ, the myelin component protein PMP22 [64]. DFT2s reduced response to IFNγ at differentiation markers is consistent with the hypothesis that DFT2 has emerged from a repair Schwann cell which has already altered its gene expression profile in response to injury, and thus shows minimal dedifferentiation.

Schwann cells can play an active role in antigen presentation, particularly following nerve injury where Schwann cells upregulate MHC-I expression from a low basal level, produce inflammatory cytokines to recruit phagocytes and stimulate proliferation [45,52,65–67]. In lieu of primary Tasmanian devil Schwann cells, we have assessed the expression of MHC-I in devil sciatic nerve sections, a tissue which is highly enriched for Schwann cells [68]. Consistent

with other species, Tasmanian devil Schwann cells express low levels of classical MHC-I (Fig 4A), a feature which has been retained by the MHC-I negative DFT1 [46] (Fig 4C), aiding in immune evasion as the tumour has spread between hosts [48]. The variable classical MHC-I expression in DFT2 tumours [28] (Fig 4E and 4G) reflects its emergence from a repair Schwann cell which has upregulated MHC-I expression in response to nerve damage, but retains the ability to modulate expression, giving rise to phenotypic diversity. Interestingly, Tasmanian devil Schwann cells express high levels of a non-classical MHC-I molecule, Saha-UK (Fig 4B). This molecule has been previously found to be expressed in DFT2 cells and some tumours [28]. Saha-UK is monomorphic [69,70] and is unlikely to result in acute allogeneic rejection of the grafted cells [71]. In addition, non-classical MHC-I molecules have been implicated in immune suppression by cancers and in maintaining immune privilege at the foetal-maternal interface [72,73]. Human Schwann cells have been shown to express a lipid presenting non-classical MHC-I molecule CD1d, which is able to interact with invariant Natural Killer T cells to modulate the immune response [74]. The function of Saha-UK is currently unknown, but the enhanced expression of a potentially immunomodulatory MHC-I molecule may aid in the emergence of transmissible tumours from this cell type.

MHC-I loss in DFT1 and several other low MHC-I tumours has been previously linked to epigenetic silencing of the antigen presentation pathway by PRC2 [48]. It has been previously shown that though MHC-I positive, DFT2_RV cells express lower levels of MHC-I than devil fibroblasts [28]. Here we show that reduced MHC-I expression in DFT2 cells relative to fibroblast cells is not driven by PRC2, and no expression of EED, a key functional subunit of PRC2, can be detected in DFT2 cells (Fig 4D). In Schwann cells, PRC2 is responsible for silencing of immature and repair Schwann cell genes such as FABP7 and L1CAM following nerve injury resulting in the redifferentiation of repair Schwann cells to a myelinating phenotype [75,76], and thus the lack of EED expression by DFT2 cells is consistent with emergence from a dedifferentiated repair Schwann cell post nerve injury [76]. This highlights a potential mechanism for the emergence of phenotypically diverse transmissible Schwann cell tumours, whereby MHC-I expression and differentiation state are coupled through PRC2 mediated gene silencing, resulting in distinct molecular phenotypes and immune evasion mechanisms despite their shared cellular origins.

In the presence of MHC-I, DFT2 cells must be utilising additional immunomodulatory mechanisms to avoid immune detection between hosts. A frequent mechanism of immune evasion in malignant Schwann cell tumours is upregulation of immune checkpoint molecule PD-L1 in response to IFNγ [77,78]. We find that DFT2 cells, similar to DFT1, upregulate PD-L1 and MHC-I following IFNγ treatment [49], highlighting a conserved immune evasion mechanism (Fig 3D). STAT3 is a transcriptional activator which is essential to the long term survival of repair Schwann cells following nerve injury [79], and has been implicated in the progression of Schwann cells to malignancy [80]. STAT3 is constitutively hyperactivated in DFT1, involved in the down-regulation of MHC-I in this tumour [81], and here we have demonstrated that DFT2 cells express similar levels of STAT3 to DFT1, despite still expressing MHC-I (Fig 3D). This may indicate that in contrast to DFT1, STAT3 is not a major driver of immune evasion or MHC-I loss in DFT2.

The TGFβ signalling pathway, and key cytokines TGFβ1 and TGFβ2 have been linked with several immunosuppressive functions [82–86], and play a crucial role in the proliferation of immature and repair Schwann cells after nerve injury, preventing myelin production and maintaining cells in an immature-like state [13,43,87]. Here, we have shown that whilst DFT1 expresses low levels of TGFβ1/2 under normal conditions, DFT2 demonstrates constitutive expression (Fig 3D). While both are upregulated by IFNγ in DFT1, their expression is unaffected in DFT2. DFT2 may have maintained high levels of TGFβ expression from its

progenitor cell, which has had the additional fitness benefit to the tumour of contributing to an immunosuppressive microenvironment.

Schwann cell tumours in humans do not frequently progress to malignancy [19], but there is some evidence that the progression of benign Schwann cell tumours to aggressive, malignant and metastatic disease is more strongly associated with emergence from less well differentiated progenitor cells [21,24,25]. Given the recent emergence and relatively low prevalence of DFT2 [7], there is not currently enough information to draw conclusions on how DFT2s progenitor cell may be influencing these characteristics in the wild, but the emergence of DFT2 from a less well differentiated Schwann cell may indicate that the tumour is more likely to be highly aggressive and metastatic, a point of concern which should be closely monitored as part of conservation efforts. Interestingly, recent studies of CTVT, a contagious cancer circulating in dogs, speculate a neural crest origin for this tumour, suggesting that the ability of contagious cancers to emerge may be linked to this lineage [88].

The emergence of parasitic Schwann cell cancers as a major pathogen in the Tasmanian devil challenges our understanding of cancer and infectious disease. The independent emergence from Schwann cells suggests that these cells may be uniquely primed for forming contagious cancers due to their inherent plasticity and adaptability to environmental cues. We have shown that the DFT1 and DFT2 progenitor cells were in different differentiation states which has directed the emergence of tumours with distinct molecular phenotypes and immune evasion mechanisms through retention of inherent properties of their progenitor cells, and have identified a potential mediator of this phenotypic diversity in PRC2. Contagious cancers are considered to be rare phenomena, but in recent years a mounting number of such pathogens have been identified, particularly in marine bivalves [89,90], indicating that cancer as a transmissible pathogenic agent may be more common in nature than previously thought. Given the significant burden cancer represents in wild populations [27], the emergence of two parasitic tumours from a single cell type in the Tasmanian devil signals the need for further monitoring of wildlife cancers and emerging wildlife pathogens more generally.

## Materials and methods

### Ethics statement

All animal procedures were performed under a Standard Operating Procedure approved by the General Manager, Natural and Cultural Heritage Division, Tasmanian Government Department of Primary Industries, Parks, Water and the Environment or under University of Tasmania Animal Ethics Committee Permit A0014976.

### Cell culture and treatments

Representative cell lines which have been previously described were used for DFT1 (DFT1_4906 [91,92]), DFT2 (DFT2_RV [6], DFT2_SN [6], DFT2_549 [28]) and fibroblasts (Fibs_Salem [92]). Cells were cultured at 35°C, 5% $CO_2$ in RMPI-1640 + glutamine (Gibco, Cat no. 61870044) supplemented with 10% heat inactivated Foetal Bovine Serum (FBS) (Gibco, Cat no. 10500–064) and 50 μg/mL penicillin/streptomycin (Gibco, Cat no. 15070063) (DFT1 and DFT2) or DMEM + glutamine (Gibco, Cat no. 10566–016) supplemented with 10% FBS (Gibco, Cat no. 10099141) and 50 μg/mL penicillin/streptomycin (Fibs_Salem). Wild type CHO cells and CHO cells stably transfected with Tasmanian Devil IFNγ [46] were cultured at 37°C, 5% $CO_2$ in DMEM/F-12 + glutamine (Gibco, Cat no. 11320033) supplemented with 10% FBS and 50 μg/mL penicillin/streptomycin. Cells were harvested using trypsin replacement enzyme TrypLE Express (Gibco, Cat no. 12605036), washed with Dulbecco's PBS

(Gibco, Cat no. 14190094) and split 1:3 (DFT1, DFT2 and Fibroblasts) or 1:10 (CHO and CHO_IFNγ) every 3–4 days.

IFNγ treatment was carried out on DFT1 and DFT2 cell lines as previously described using recombinant Tasmanian devil IFNγ [46]. Cell culture medium from confluent CHO_IFNγ cells was filtered using a 0.2 μM filter to remove cells and debris, then mixed in a 1:1 ratio with DFT culture media. Media from untransfected CHO cells was used as a control. IFNγ containing media was added to DFT1 and DFT2 cells at ~80% confluency for 16 hours. Cells were harvested and washed after 16 hours and immediately stained for cell surface $\beta_2$_m or snap frozen using dry ice for RNA extraction and gene expression analysis.

## Proteomics

DFT1_4906, DFT2_RV and Fibs_Salem cells (n = 3) were lysed in buffer containing 1% sodium deoxycholate (SDC), 7.5 mM TCEP and 50mM TEAB. The lysate was subjected to probe sonication to sheer DNA and clarified by centrifugation. The disulphide bonds of proteins in the lysates were reduced by incubating with 10 mM TCEP at 60˚C for 15 min followed by alkylation by 40 mM chloracetamide for 15 min at RT. The proteins were then digested using trypsin overnight at 37˚C. SDC was removed by acid precipitation using trifluoroacetic acid followed by centrifugation for 15 min at 16060xg. The resultant peptides were fractionated using a 10 cm x 4.6 mm Chromolith C18 column running a mobile phase consisting of buffer A (aqueous 0.1% TFA) and buffer B (80% ACN/0.1% TFA). The peptides were eluted using a linear gradient of 5 to 40% buffer B over 30 min collecting 38 fractions which were concatenated into 7 pools before centrifugal evaporation. Samples were reconstituted using 0.1% formic acid and analysed using Thermo Scientific Q-Exactive Plus with following settings: survey scan (375–1800 m/z) at 70,000 resolution (m/z 200) accumulation target: 5e5 ions, maximum injection time: 120 ms. The 12 most intense precursors were fragmented using following settings: higher-energy collisional dissociation (HCD) at 27%, injection time of 120 ms, 35000 resolution and target of $2 \times 10^5$ counts. An isolation width of 1.8 m/z was applied and underfill ratio was set to 1% and dynamic exclusion to 15 sec. Data obtained was searched against the Tasmanian devil protein database obtained from NCBI (20170301) using MaxQuant software with a false discovery rate set to 1% with following parameters: enzyme: trypsin, variable modifications: oxidation (M), Acetyl (Protein N-Term), fixed modifications: Carbidomethyl (C) [93]. Label Free Quantitation (LFQ) was performed in Perseus[94], and only proteins present in all three replicates of any one sample were considered for quantitation. The mass spectrometry proteomics data have been deposited to the ProteomeXchange Consortium via the PRIDE partner repository with the dataset identifier PXD021784.

Pairwise comparisons were performed using pairwise T-tests between the quantified proteins in DFT1_4906 and Fibs_Salem, DFT2_RV and Fibs_Salem and DFT1_4906 and DFT2_RV (S1–S3 Tables). Pairwise comparisons were corrected for multiple testing artefacts by controlling the false discovery rate using a q-value cut off of < 0.05, where only proteins with a q-value of < 0.05 were considered for further analysis [95,96]. T-test p-values of < 0.05 indicate proteins which are significantly differentially expressed between two cell lines. RefSeq protein IDs were converted to Official Gene IDs using a combination of Uniprot and Ensembl databases, and where no annotation was present in the genome proteins were manually annotated based on homology to other species. Differentially expressed proteins were ranked based on fold change values and proteins which were significantly overexpressed at least 2 fold in a given cell line were analysed as a ranked list using the g:Profiler g:GOSt web application [97,98] (S4–S7 Tables). The database used in this analysis is the Gene Ontology (GO) Consortium Biological Process (GO:BP) database [99,100]. Analysis was run using a human

background to avoid loss of information due to poor functional annotation of the Tasmanian devil genome. Functional analysis used a significance cut off of p-value < 0.05, where the p-value represents the significance of pathway enrichment corrected for multiple testing using the g:SCS algorithm, a custom made algorithm which considers hierarchical nature of functional analysis terms which has been validated [97,98]. Nervous system associated biological processes were manually compiled and their significance visualised as a bar chart.

## Flow cytometry

Cell surface $\beta_2$-m was determined by flow cytometry as previously described [46] using a validated Tasmanian devil specific antibody (B$_2$-m_13-34-48). Fluorescence was determined using a Merck Guava easyCyte benchtop system. Data was analysed using Guava InCyte 3.3 software.

## Reverse transcription and gene expression analysis

RNA was extracted from frozen cell pellets using Nucleospin RNA mini kit (Macherey-Nagal, Cat no. 740955.50) following the manufacturer's instructions. RNA was eluted and the concentration and purity determined using a Nanodrop 2000. RNA was immediately converted to cDNA using Revertaid RT Reverse transcription kit (Thermofisher Scientific, cat. no. #K1691) with 1000 ng of RNA, 20 μM Oligo (dT)$_{15}$ primer and 0.5 mM dNTPs. The reaction was heated to 65°C for 5 minutes and immediately chilled on ice. 200 U Revertaid RT reverse transcriptase in 1 X Revertaid RT buffer was added and incubated at 42°C for 30 minutes, followed by 80°C for 5 minutes.

Gene expression analysis was performed by Quantitative Reverse Transcription-PCR (qRT-PCR). Primers were designed from the DEVIL7.0 assembly of the devil genome (GCA_000189315.1) accessed via Ensembl. A full list of primers and conditions can be found in S8 Table at https://doi.org/10.5258/SOTON/D1975. qRT-PCR was performed using 250 ng of cDNA, 0.25 μM of forward and reverse primers and 1X PrecisionPlus qPCR Master Mix with SYBR green and ROX (PrimerDesign, PrecisionPLUS-R-SY) on a BrightWhite real-time PCR plastic qPCR plate (PrimerDesign, BW-FAST). qRT-PCR experiments were run on a StepOne Plus real-time PCR system (Applied Biosciences) on a standard protocol with a melt curve to assess amplicon purity.

All qRT-PCR plates were run alongside a standard curve for each primer set, consisting of a high expressing control cell line cDNA serially diluted 1:2 or 1:5 with negative controls. All standard curves had $R^2$ values of >0.98 and efficiencies of 80–120%. All samples and standard curves were plated in duplicate. Melt curves were visually inspected following amplification to confirm amplification of the expected gene fragment, and any samples not amplifying the expected fragment were removed from further analysis. Gene of interest expression was quantified using the relative standard curve method, relative to the expression of RPL13A which has been previously validated as a housekeeping gene in the Tasmanian devil [46]. For Fig 2A, 2C and 2D, T-tests were performed on the average relative expression of each gene in DFT1 and DFT2 with and without IFNγ treatment. These experiments involve a small number of co-regulated genes, which were selected for analysis *a priori* based on the wide literature surrounding Schwann cell responses to IFNγ, and thus statistical correction for multiple comparisons is not necessary [101] or standard practice for the analysis of such experiments [102,103], due to the introduction of an unacceptable level of Type II errors, particularly when applied to co-regulated gene sets [104,105]. For Fig 2B, two-way ANOVA was performed on average expression values across three biological replicates to compare the effects of cell line and treatment on gene expression.

Additional gene expression analysis was performed by RT-PCR using 250 ng of cDNA, 0.2 μM of forward and reverse primers and 2X PCR Master Mix (Thermofisher, Cat. no. #K0171) over 29 cycles. RPL13A is used as a housekeeping control as above. PCR amplicons were separated by gel electrophoresis using a 1.2% agarose gel containing 1 X GelRed Nucleic acid gel stain and visualised post-electrophoresis using a Syngene G:BOX. Biological IFNγ treatment replicates for genes analysed by RT-PCR are presented in S1 Fig and S2 Fig.

## Immunohistochemistry

Wild Tasmanian devils were either trapped or found dead from road trauma or other causes. Tissue biopsies were either collected post mortem or from live devils that were subsequently released.

Sciatic nerve, DFT1, and DFT2 tissue samples were fixed in 10% (vol/vol) PBS-buffered formalin solution for 2 to 4 days and embedded in paraffin blocks. Sections were cut using a microtome at a thickness between 4–8 μm and mounted onto coated slides.

Sections were deparaffinized in xylene and rehydrated through graded alcohol. Antigen retrieval was performed by water bath (95°C) in citrate buffer solution (10 mM citric acid, 0.05% Tween20, pH 6) for 40 min followed by cooling for 15 min. Slides were incubated with 3% $H_2O_2$ (Sigma Aldrich) for 10 min, followed by incubation with 10% (vol/vol) goat serum in PBS for 30 min. Sections were incubated at 4°C overnight with primary antibody. Antibodies against all three Tasmanian devil classical MHC class I heavy chains (Saha-UA/UB/UC, 15-25-8) and the non-classical MHC class I molecule Saha-UK (15-29-1) were used neat, as previously described [28]. Primary antibody binding was detected using peroxidase-coupled secondary antibody (Dako REAL EnVision HRP); slides were incubated with HRP for 30 min followed by colour development with DAB chromogen for 5 min (Dako REAL DAB for sciatic nerve and Vector ImmPACT DAB for DFT1 and DFT2). Sections were counterstained with haematoxylin (Vector haematoxylin (Gill's Formula) for sciatic nerve, Sigma-Aldrich haematoxylin (Harris' Formula, Modified) for DFT1 and DFT2) for 4 min, differentiated in 2% (vol/vol) acetic acid and blued in 0.2% (vol/vol) ammoniated water. Sections were dehydrated through graded alcohol, transferred to xylene and covered using Omnimount Histological Mounting Medium (National Diagnostics, HS-110). Images were captured using the Nikon Eclipse 400 microscope, Retiga 2000R camera and Q-capture pro 7 computer software. Scale bars were added to images using ImageJ software.

## Western blot

Cell lysate was generated from frozen cell pellets by resuspending in lysis buffer with 1% NP-40, and incubation at 4°C for 30 mins. Lysates were clarified by centrifugation for 10 mins at 11000 RCF at 4. Protein concentration was determined by Bradfords assay using Pierce Coomassie (Bradford) Protein Assay Kit (Thermofisher Scientific, Cat no. 23200). Proteins were separated by SDS-PAGE using 12% SDS resolving gel and 4% stacking gel (National Diagnostics Protogel 30% (EC-890), Resolving buffer (EC-892), Stacking buffer (EC-893)). 30 μg of protein per sample was denatured with β-mercaptoethanol at 95°C for 10 mins, then loaded into the gel along with a protein ladder (Geneflow, Cat no. S6-0024). Protein was transferred to a nitrocellulose membrane and blocked with 5% milk powder (Marvel) in TBST, followed by incubation with primary Ab diluted in TBST (EED (Sigma-Aldrich, 09–774, 1:2000), Beta-Actin (Proteintech, 2D4H5, 1:50,000), Saha-UA/UB/UC (15-25-8, neat hybridoma supernatant)) overnight at 4°C. Membranes were washed in TBST and incubated with secondary antibody diluted 1:15000 in 5% milk (LiCor, goat anti-mouse 680 (925–68070), goat anti-rabbit 680 (925–68071) at room temperature for 1 hr, washed in TBST and imaged on a Li-cor scanner.

## Supporting information

**S1 Fig. Agarose gels showing independent biological replicates of MHC-I modulation by IFNγ in DFT1 and DFT2.** Agarose gels showing RT-PCR results for MHC class I associated genes on three independent biological replicates of DFT1_4906 and DFT2_RV, DFT2_SN and DFT2_549 treated with recombinant devil IFNγ in vitro. C indicates control cells grown in normal cell culture media, γ indicates cells treated for 16 hours with recombinant devil IFNγ. ND indicates a no cDNA negative control. L indicates a DNA ladder containing fragments of known size. Key molecular weights are indicated by blue arrowheads (300 bp) and red arrows (200 bp). RPL13A is a housekeeping gene used to control for the amount of cDNA in each PCR reaction. Related to Fig 3C.
(TIF)

**S2 Fig. Agarose gels showing independent biological replicates of immune gene modulation by IFNγ in DFT1 and DFT2.** Agarose gels showing RT-PCR results for a panel of genes associated with immune function on three independent biological replicates of DFT1_4906 and DFT2_RV cell lines treated with recombinant devil IFNγ in vitro. C indicates control cells grown in normal cell culture media, γ indicates cells treated for 16 hours with recombinant devil IFNγ. ND indicates a no cDNA negative control. L indicates a DNA ladder containing fragments of known size. Key molecular weights are indicated by blue arrowheads (300 bp) and red arrows (200 bp). RPL13A is a housekeeping gene used to control for the amount of cDNA in each PCR reaction. Related to Fig 3E.
(TIF)

**S3 Fig. Secondary antibody only controls of histological sections stained by IHC.** All images taken at 200X magnification. Scale bars indicate 100 µM. Black arrows indicate tumour cells, and black arrowheads indicate Schwann cells which are shown at a higher magnification in Fig 4. Blue staining indicates nuclei counterstained with haematoxylin. A) Tasmanian devil sciatic nerve, B) DFT1 tumour (Falestinya T1), C) DFT2 tumour (812 T1), D) DFT2 tumour (547 T1). Related to Fig 4.
(TIF)

**S1 Table. Pairwise comparisons of differentially expressed proteins between DFT1_4906 vs Fibs_Salem.** Each row represents a single protein and Official Gene Identifiers are provided in Column AC. Columns A-I represent quantification data for each protein in triplicate samples for each cell line. Column X (N: Student's T-test Difference) indicates Log2 Fold Change in expression, where positive values indicate an overexpression and negative values indicate an underexpression as calculated by pairwise comparisons. Column V (N: -Log Student's T-test p-value) indicates the Log2 P-value of each pairwise comparison, where a value of approximately 1.3 corresponds to a significant result of $p < 0.05$. Related to Fig 1A–1D.
(XLSX)

**S2 Table. Pairwise comparisons of differentially expressed proteins between DFT2_RV vs Fibs_Salem.** Each row represents a single protein and Official Gene Identifiers are provided in Column AC. Columns A-I represent quantification data for each protein in triplicate samples for each cell line. Column X (N: Student's T-test Difference) indicates Log2 Fold Change in expression, where positive values indicate an overexpression and negative values indicate an underexpression as calculated by pairwise comparisons. Column V (N: -Log Student's T-test p-value) indicates the Log2 P-value of each pairwise comparison, where a value of approximately 1.3 corresponds to a significant result of $p < 0.05$. Related to Fig 1A–1D.
(XLSX)

**S3 Table. Pairwise comparisons of differentially expressed proteins between DFT2_RV vs DFT1_4906.** Each row represents a single protein and Official Gene Identifiers are provided in Column AC. Columns A-I represent quantification data for each protein in triplicate samples for each cell line. Column X (N: Student's T-test Difference) indicates Log2 Fold Change in expression, where positive values indicate an overexpression and negative values indicate an underexpression as calculated by pairwise comparisons. Column V (N: -Log Student's T-test p-value) indicates the Log2 P-value of each pairwise comparison, where a value of approximately 1.3 corresponds to a significant result of $p < 0.05$. Related to Fig 1A–1D. (XLSX)

**S4 Table. Gene Ontology: Biological Process (GO:BP) analysis for proteins overexpressed at least 2 fold in DFT1_4906 vs Fibs_Salem.** Analysis completed as described in Materials and Methods using gProfiler g:GOSt web application [97,98]. Column D (adjusted_p_value) indicates the significance of pathway enrichment in the dataset corrected for multiple testing using the g:SCS algorithm, a custom made algorithm which considers hierarchical nature of functional analysis terms which has been validated [97,98]. Column H (intersection_size) indicates the number of proteins in the dataset found to be associated with a given biological process. Column J (intersections) lists the Official Gene ID for all intersecting proteins. Related to Fig 1E. (XLSX)

**S5 Table. Gene Ontology: Biological Process (GO:BP) analysis for proteins overexpressed at least 2 fold in DFT2_RV vs Fibs_Salem.** Analysis completed as described in Materials and Methods using gProfiler g:GOSt web application [97,98]. Column D (adjusted_p_value) indicates the significance of pathway enrichment in the dataset corrected for multiple testing using the g:SCS algorithm, a custom made algorithm which considers hierarchical nature of functional analysis terms which has been validated [97,98]. Column H (intersection_size) indicates the number of proteins in the dataset found to be associated with a given biological process. Column J (intersections) lists the Official Gene ID for all intersecting proteins. Related to Fig 1E. (XLSX)

**S6 Table. Gene Ontology: Biological Process (GO:BP) analysis for proteins overexpressed at least 2 fold in DFT2_RV vs DFT1_4906.** Analysis completed as described in Materials and Methods using gProfiler g:GOSt web application [97,98]. Column D (adjusted_p_value) indicates the significance of pathway enrichment in the dataset corrected for multiple testing using the g:SCS algorithm, a custom made algorithm which considers hierarchical nature of functional analysis terms which has been validated [97,98]. Column H (intersection_size) indicates the number of proteins in the dataset found to be associated with a given biological process. Column J (intersections) lists the Official Gene ID for all intersecting proteins. Related to Fig 1E. (XLSX)

**S7 Table. Gene Ontology: Biological Process (GO:BP) analysis for proteins overexpressed at least 2 fold in DFT1_4906 vs DFT2_RV.** Analysis completed as described in Materials and Methods using gProfiler g:GOSt web application [97,98]. Column D (adjusted_p_value) indicates the significance of pathway enrichment in the dataset corrected for multiple testing using the g:SCS algorithm, a custom made algorithm which considers hierarchical nature of functional analysis terms which has been validated [97,98]. Column H (intersection_size) indicates the number of proteins in the dataset found to be associated with a given biological process. Column J (intersections) lists the Official Gene ID for all intersecting proteins. Related to

.
(XLSX)

**S8 Table. Full list of primers and PCR conditions used to generate data presented in Figs 2 and 3.**
(DOCX)

**S9 Table. Quantitative PCR data used to generate Fig 2.**
(XLSX)

## Acknowledgments

The authors thank Gregory Woods, Alexandre Kreiss and Ruth Pye from the Menzies Research Institute for provision of Tasmanian devil tissue samples and valuable discussion.

## Author Contributions

**Conceptualization:** Rachel S. Owen, Anthony W. Purcell, Hannah V. Siddle.

**Formal analysis:** Rachel S. Owen, Sri H. Ramarathinam, Alistair Bailey, Annalisa Gastaldello, Paul J. Skipp, Hannah V. Siddle.

**Funding acquisition:** Anthony W. Purcell, Hannah V. Siddle.

**Investigation:** Rachel S. Owen, Sri H. Ramarathinam, Alistair Bailey, Annalisa Gastaldello, Kathryn Hussey.

**Supervision:** Paul J. Skipp, Anthony W. Purcell.

**Writing – original draft:** Rachel S. Owen, Hannah V. Siddle.

**Writing – review & editing:** Rachel S. Owen, Sri H. Ramarathinam, Alistair Bailey, Annalisa Gastaldello, Kathryn Hussey, Paul J. Skipp, Anthony W. Purcell, Hannah V. Siddle.

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
