## [Decision Letter · Decision Letter 0]

8 Sep 2021

Dear Dr. Siddle,

Thank you very much for submitting your manuscript "The differentiation state of the Schwann cell progenitor drives phenotypic variation between two contagious cancers" for consideration at PLOS Pathogens. As with all papers reviewed by the journal, your manuscript was reviewed by members of the editorial board and by three independent reviewers, including two (Reviewers 1 and 3) that had previously reviewed a version. You will see that the reviewers are still mixed in their appraisal of the paper. All reviewers would like to see the paper pitched in more general terms rather than so focused so as to only appeal to a very limited audience. reviewer 2 makes specific recommendations in this regard. Reviewer 3 is still quite negative about the accuracy and impact of your findings. They raise legitimate concerns about the statistics used and whether they accounted for the multiple testing. We will require you to address these criticisms in your revision, with specific mention of the types of safeguards used to guard against multiple testing artifacts. Reviewer 3 is also concerned that the level of impact of these findings may not rise to the level that we expect of papers at PLOS Pathogens, especially in light of recent findings. However, based on comments of the other two reviewers, who are also experts in this field (like Reviewer #3), editorially we do not believe this would be a significant impediment to ultimate acceptance of your paper here, if you can satisfactorily address the other comments raised. We do encourage you to cite all prior literature and draw your attention to PLOS policy of scoop protection. Therefore, in light of the reviews (below this email), we would like to invite the resubmission of a significantly-revised version that takes into account the reviewers' comments.

We cannot make any decision about publication until we have seen the revised manuscript and your response to the reviewers' comments. Your revised manuscript may be sent to reviewers for further evaluation if needed.

Sincerely,

Harmit S. Malik

Guest Editor

PLOS Pathogens

Marco Vignuzzi

Section Editor

PLOS Pathogens

Kasturi Haldar

Editor-in-Chief

PLOS Pathogens

orcid.org/0000-0001-5065-158X

Michael Malim

Editor-in-Chief

PLOS Pathogens

orcid.org/0000-0002-7699-2064

Reviewer's Responses to Questions

**Part I - Summary**

Reviewer #1: In the Owen et al manuscript, the authors describe the Schwann cell origins of both lineages of Tasmanian devil facial tumour disease (DFT1 and DFT2). It had previously been shown that both lineages arose from Schwann cells, but this report goes further to show that one lineage is more differentiated than the other (DFT1 is more differentiated). Notably, this has an effect on the immune evasion strategies of these cancers, as the differentiation state affects MHC-I expression level. The authors have answered all of the reviewers' questions, removing a problematic section and adding significantly improved discussion and analysis of proteomic data.

Reviewer #2: This manuscript compares the differentiation state of Schwann cells that gave rise to two independent transmissible cancers in Tasmanian devils - DFT1 (the original lineage, described in the 1990s) and DFT2 (described in 2016). Although the gross morphology of tumors on the face of Tasmanian devils is quite similar, DFT1 and DFT2 present differently in terms of histology, and genomic differences have been identified between the two tumors. The cells of DFT1 and DFT2 also have differing levels of MHC expression, as well as responses to IFN-gamma treatment. The experiments herein, primarily a proteomics analysis, show that DFT2 emerged from a less differentiated Schwann cell than DFT1. I have not seen a previous version of the MS and note that the authors have done a good job responding to previous comments and removing less well-supported data. My main concern is that this manuscript is quite specific relating to Tasmanian devils and DFTD, and as written, may not be of general interest to the broad readership of PLoS Pathogens. As such, my general recommendation is to introduce the manuscript in a more broad way (which is done to some extent in the discussion) as to the implications for understanding the differentiation state of Schwann cell tumors. The authors mention the plasticity of Schwann cells, and perhaps a more general discussion as to how this relates to them becoming cancerous would enhance the introduction.

Reviewer #3: In the revised version of their paper, Owen et al document (1) whole-proteome differences in protein expression between representative DFT1 and DFT2 cell lines, (2) alterations in transcript abundance for selected genes in representative DFT1 and DFT2 cell lines cultured in the presence and absence of recombinant IFNg, (3) differences in the magnitude of upregulation of cell-surface B2M following IFNg treatment among several DFT2 cell lines as well as a DFT1 cell line, and (4) using immunohistochemistry, assess expression of SAHA UABC and UK in Tasmanian devil sciatic nerve and in representative DFT1 and DFT2 tumour biopsies.

In the revision, the authors have performed replicates of the qPCR experiments, and these are now considerably more robust. However, I struggle to follow a coherent story in the paper. Much of the work repeats findings already in the published literature (Patchett et al 2019, Ong et al 2021, Flies et al 2016, Siddle et al 2013, Caldwell et al 2018). The main new contribution is the proposal that DFT1 and DFT2 arose from different progenitor cell states. However, this was already hinted at in Patchett et al 2019, and, in my opinion, despite the authors’ rebuttal, I do not believe that these cancers’ progenitor cell states can be inferred from the presented work. The PCR data presented in figures 2C, 2D, 3C, 3D, S1, S2 detect subtle changes and statistical significance level does not seem to have been corrected for multiple testing. I am left unsure that differences are significant and reproducible. Furthermore, fig 3B shows that there is significant variation in cell surface B2M level among DFT2 cell lines; this makes me wonder how many of the findings from the rest of the paper are specific to the single representative cell lines selected for DFT1 and DFT2.

Overall, I don’t find the results a compelling-enough contribution to the field to warrant publication in PLOS Pathogens.

**Part II – Major Issues: Key Experiments Required for Acceptance**

Reviewer #1: (No Response)

Reviewer #2: Major issues seem to have been addressed by previous reviewers, and the authors removed the questionable data sets from the manuscript. I do not see any additional major issues of concern.

Reviewer #3: (No Response)

**Part III – Minor Issues: Editorial and Data Presentation Modifications**

Reviewer #1: (No Response)

Reviewer #2: I could nitpick, but the manuscript is generally well-written. However, as noted above, the manuscript as written is for a specialized audience, and a more general discussion of the relevance of the data for cancer in general would broaden the scope of the paper appropriately for PLoS Pathogens.

Reviewer #3: (No Response)

PLOS authors have the option to publish the peer review history of their article (what does this mean?). If published, this will include your full peer review and any attached files.

Reviewer #1: No

Reviewer #2: No

Reviewer #3: No
---

## [Editor Report · Decision Letter 1]

13 Oct 2021

Dear Dr. Siddle,

We are pleased to inform you that your manuscript 'The differentiation state of the Schwann cell progenitor drives phenotypic variation between two contagious cancers' has been provisionally accepted for publication in PLOS Pathogens. Thank you for your thoughtful revisions.

Best regards,

Harmit S. Malik

Guest Editor

PLOS Pathogens

Marco Vignuzzi

Section Editor

PLOS Pathogens

Kasturi Haldar

Editor-in-Chief

PLOS Pathogens

orcid.org/0000-0001-5065-158X

Michael Malim

Editor-in-Chief

PLOS Pathogens

orcid.org/0000-0002-7699-2064
---

## [Editor Report · Acceptance letter]

9 Nov 2021

Dear Dr. Siddle,

We are delighted to inform you that your manuscript, "The differentiation state of the Schwann cell progenitor drives phenotypic variation between two contagious cancers," has been formally accepted for publication in PLOS Pathogens.

Best regards,

Kasturi Haldar

Editor-in-Chief

PLOS Pathogens

orcid.org/0000-0001-5065-158X

Michael Malim

Editor-in-Chief

PLOS Pathogens

orcid.org/0000-0002-7699-2064